# My Practice of Re-Patterning My Art

## Pramila Vasudevan

Aniccha Arts, St. Paul, MN 55116, USA; pramila@aniccha.org

**Abstract:** This essay shares the various ways in which my socio-political context and health background has impacted my journey as an artist and culture worker through my work with the Aniccha Arts collaborative in the Twin Cities. I would like to share how my (un/re)learnings have materialized into different movement textures of togetherness over the years. I describe how I arrived at creating the current movement-based project, Prairie | Concrete, and the questions that I am asking as a path forward.

**Keywords:** repattern; caste; site-responsive; decolonize; land-becoming; togetherness; reciprocity; embodied



## 1. Struggles and Complicities in My Artist Journey

My goal in this essay is to reveal the struggles and complicities of my journey as a transdisciplinary artist and dance maker over the last eleven years. This essay shares the various ways in which my socio-political context and health background has impacted my journey as an artist and culture worker through my work with the Aniccha Arts[1] collaborative in the Twin Cities. I would like to share how my (un/re)learnings[2] have materialized into different movement textures of togetherness over the years. I describe how I arrived at creating the current movement-based project, Prairie | Concrete[3] (Vasudevan 2023), and the questions that I am asking as a path forward.

The ongoing racial reckoning that was at the surface of the Twin Cities' socio-political climate during the pandemic, with George Floyd's murder three blocks from where I worked at Pillsbury House Theatre (PHT)[4] for eleven years, has led me to actively search for action-oriented paths to be in solidarity with the Black Lives Matter movement. As a South Asian American cultural worker and artist, I have felt a responsibility to confront my own caste[5] privilege and how anti-blackness narratives are present in that social hierarchy within my body, artistic, spiritual, and daily living.

I have been trained in Bharatanatyam, a form of classical Indian dance that originated with the inherited courtesan dance communities (an oppressed caste). This form was appropriated and sanitized by "dominant caste" Indians and exported across the world. There is a violence in this history, and I am complicit; every gesture and foot rhythm that benefits me in calling my history is also an act of violence towards these communities who were not permitted to practice, teach, or share their form. This Brahmanistic mechanism of oppression is parallel to white supremacy and it fits hand in glove within the global cultural complex. In 2012, I was a lead artist in creating In Habit: Living Patterns[6] (Vasudevan 2012) (Figure 1), an all-night, outdoor project under a bridge that took a postmodern approach to using the core knowledge structures from Bharatanatyam in its construction and expression. While I was interested in asking questions about collective daily patterns, I failed to see how the actual dance patterns in this project played into the global cultural power complex at that time. I also began to learn that the pursuit of Indian dance cultivated from contexts of Brahmanism within the American dance landscape brings about a pressure for me to mold my embodied history into exotic performance material, which fits within the orientalist paradigm and thereby confines my dance world to a play space in the comfort of a theater for the white imagination and gaze.

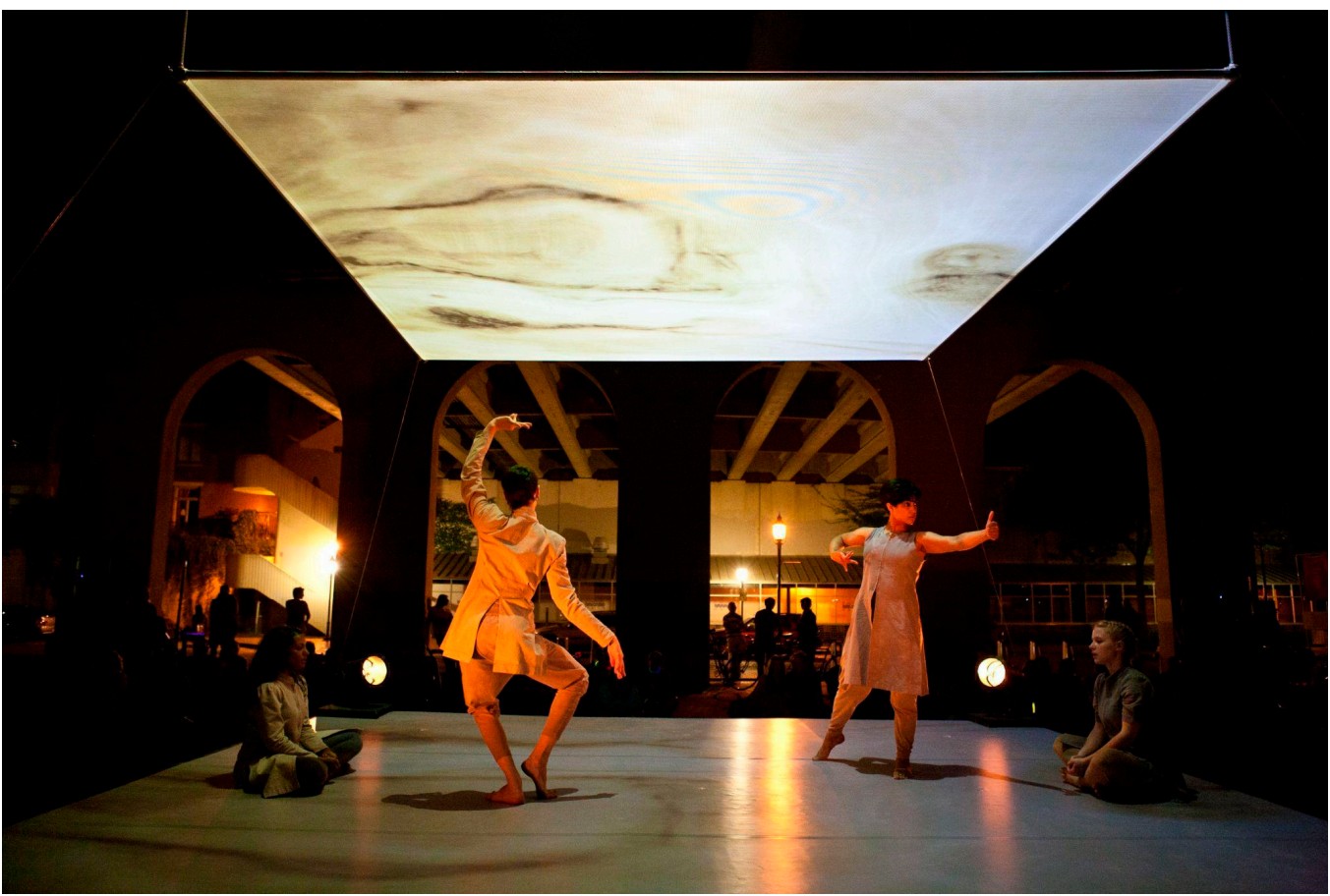

**Figure 1.** In Habit: Living Patterns, 2012 Northern Spark Festival photo by Nancy Wong.

How can I agitate this power dynamic through a repatterning of my modality of dance making? What does an anti-caste dance form look like? How do I reckon with being a South Asian American culture worker on stolen Dakota land? I will likely spend the rest of my life searching for embodied paths that engage with these questions in depth.

My work at Pillsbury House Theatre, where I facilitated classes for individuals from ages 16+ months to 80+ years, representing vulnerable communities from all walks of life, and my work at Upstream Arts[7], co-teaching social and communication skills to people with disabilities through the arts, have provided me with insight into the Twin Cities' vulnerable community fabric. These experiences have led me to open up my projects, really think about who is invited to perform in the projects that I initiate, and also rethink my values of how I arrive at a performance aesthetic. The look and feel of my own movement as a dancer presents as a texture based on Bharatanatyam, contemporary Indian dance, and somatic-based practices. Most of my collaborators do not share this movement background, and so the aesthetics of my projects are particular to the ideas and people in those works at those sites. I often collaborate with large groups of people (twenty to sixty performers) who have diverse layers of performance experience and people who are members of historically marginalized communities. I use improvisation, task-based choreographic structures, and devised choreographic tools to generate singular worlds in which movement, sound, and visuality merge into a trans-disciplinary aesthetic. The visible textures of movement are a dynamic set of behavioral patterns from the pedestrian movements of daily life offered by collaborators and elicit connection/resonance across a range of performative skillsets, presenting aesthetic patterns of our togetherness in that time and place. I continue to expand upon this way of making and cultivate ecologies of reciprocity.

My artistic practice is motivated by layers of experience. Some are personal, as a South Asian, Tamil American, third-culture kid living/working on stolen Dakota land

with a liminal sense of identity. Some are relational, where I seek to break away from settler colonialist prescriptions/ontologies to create holistic alternatives through a direct alignment/engagement with the complexities of the land and its history. I am interested in leaning into the ways that the land that we reside on forms us in terms of our physical responses to the place. Most of my work is site-responsive, so I am always thinking about the land, its materials, the people who have inhabited the site, and people who have been displaced. I often engage non-traditional sites for my performances, including parking ramps, city bridges, parkways, and wild outdoor spaces. My disciplined background in classical and contemporary Indian dance has led me to value a deep process, while, at the same time, making me sensitive to the traditional hierarchies that present opportunities for disruption.

As demonstrated by projects such as Every Other[8] (Vasudevan 2015) (Figure 2) and Census[9] (Vasudevan et al. 2016) (Figure 3), I have been interested in how people from historically marginalized communities could inhabit vast spaces. In this project, the act of activating five hundred feet of public space for one night by our team of sixty five performers from underrepresented communities might conceptually register as a powerful gesture. However, I soon realized that there is an inherent conflict between the social justice issue aligning with the 'occupy movement,' versus decolonizing our ways on stolen land and building a relationship with that place. Yet, in these two projects, intersectionality was a core motivation. Systematic injustice cannot be adequately addressed without acknowledging the many layers of identity that interact on multiple overlapping levels. Alongside the crucial work of resistance, I believe that constructive creation—imagining what is possible and making it visible—is key. My work is often large scale and facilitates multiple perspectives and choreographies playing out simultaneously, commenting on complex interrelations, as opposed to a single "universal" idea.

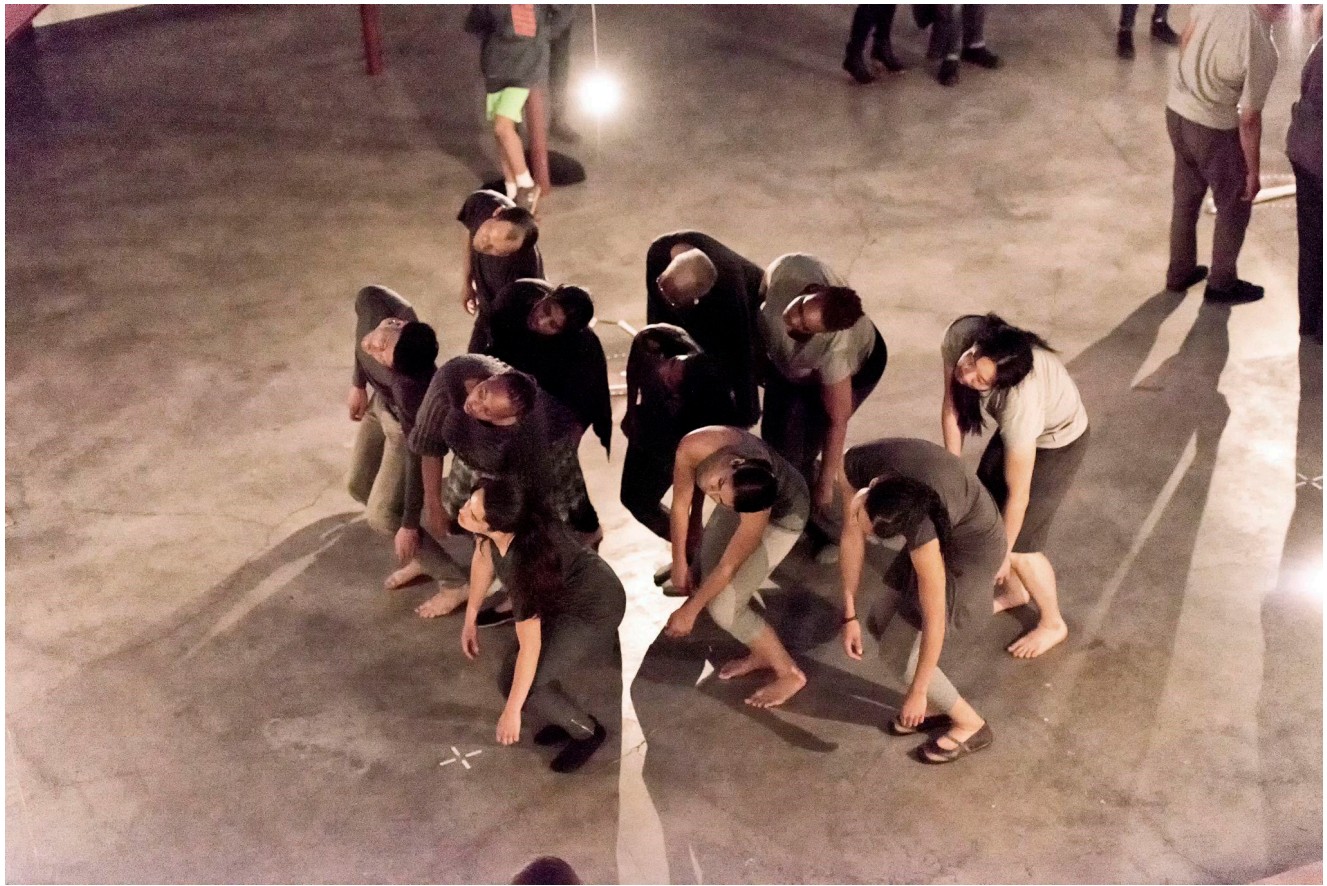

**Figure 2.** Every Other, 2015, Grain Belt Bottling House, Minneapolis, MN photo by Alice Gebura.

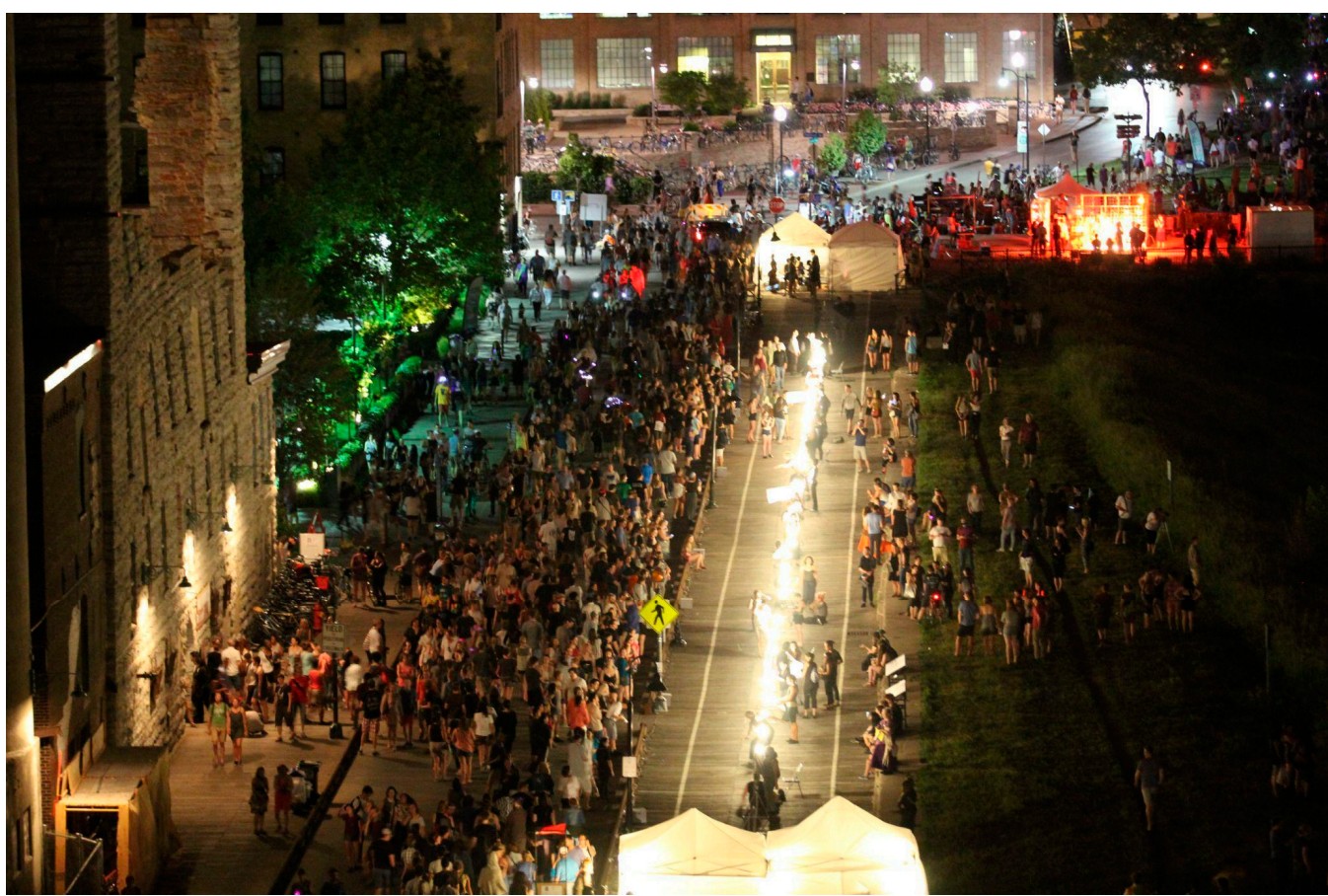

**Figure 3.** Census, 2016, Minneapolis, MN photo by Jaffa Aharanov.

## 2. Towards the Restoration of Personal Health and Artist Practice

During the COVID-19 pandemic, I was navigating my second encounter with cancer. Many medications filled me with toxicities that caused painful, frustrating side effects. While I have found some ways of reducing the side effects through diet and exercise, I know my body is permanently changed and my path to recovery has been slow. As I regain energy, I believe there is an opportunity to channel the challenges of cancer into a new chapter of my artistic practice and career. What is this newly altered body, mind, emotional state, and perspective that I have? How do I relearn how to be an artist and community member in this new reality, both on an internal level (given cancer) and on an external level (given the pandemic-wrought landscape that I am reentering as I recover)?

Two developments that have contributed significantly to my recovery included moving to a new home/neighborhood in St. Paul, MN, and starting a gardening practice. Increasingly, I have been thinking about how my expanding community in St. Paul and the lessons I am learning from plants could inform how I think about performance practice, career, and community engagement moving forward. This has led me to think about care, how it is radical and necessary, and how our communities can mend ourselves and the land. My daily practice is guided by my reckoning with the way that cancer has altered my life and has moved me towards finding paths of recovery through embodied movement and its connection to plant cycles and growing practices. I want to learn from the forests, urban farms, and city parks—how to hold the space for communities of color outdoors and root ourselves in a perspective of health that green spaces can offer.

One of my last major projects before the pandemic, Parking Ramp Project[10] (Vasudevan 2018) (Figure 4), was created inside and in response to a medical facility's parking structure. This piece engaged with harsh conditions: concrete, car fumes, and a structure that is home to nothing. While we created a beautifully embodied performance installation and

temporary community there, the project was physically intense for the participants. One day, I noticed a tiny plant that was pushing its way through the cracks in the concrete of the parking structure. Even in these harsh conditions, something organic was able to grow! I was amazed all over again by the resiliency of nature and the implications for us as humans—what our bodies can learn from plants that can seem so small and fragile, yet are able to push through concrete, insisting on life.

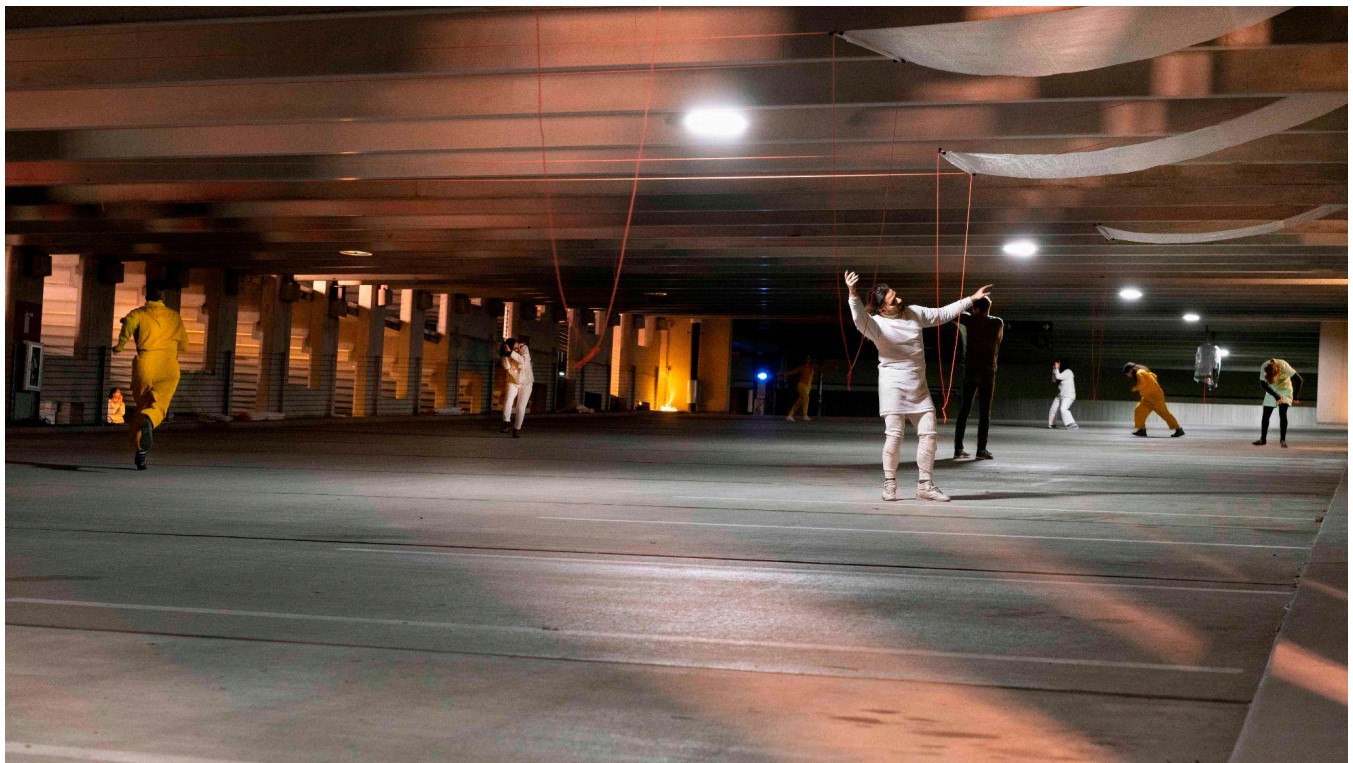

**Figure 4.** Parking Ramp Project, 2018, Health Partners Parking Ramp, Bloomington MN photo by Zoe Cinel.

Plants have a way of networking and forming habitats for other beings who depend on them. How can I, as a dance maker, learn to cultivate spaces of engagement with others inspired by the various ecologies around us? How can we move from land-owning to land-becoming? What can we learn from plant cycles and growing practices? I am also curious about how the cultivation of plants can facilitate how plants might inform human interaction and interweaving, and how art-making might influence plant growth. What are the barriers that keep people from a reciprocal relationship with nature and with each other? I want to create moments of togetherness with the public, deep engagement which will motivate us to unsettle our ways and actively teach ourselves about possibilities for an urban ecology that is not rooted in competition or capital, but in nurturing and reciprocity. How can we transform our ways and discover alternative modes of being together?

Unlike a project taking place in a theater or museum—spaces constructed with concrete that intentionally create "blank slates" for displaced imagination—my latest project, Prairie | Concrete[11] (Vasudevan 2023) (Figure 5), happens out in the open, where the sites and the surrounding communities directly contribute to the process of embodied imagining.

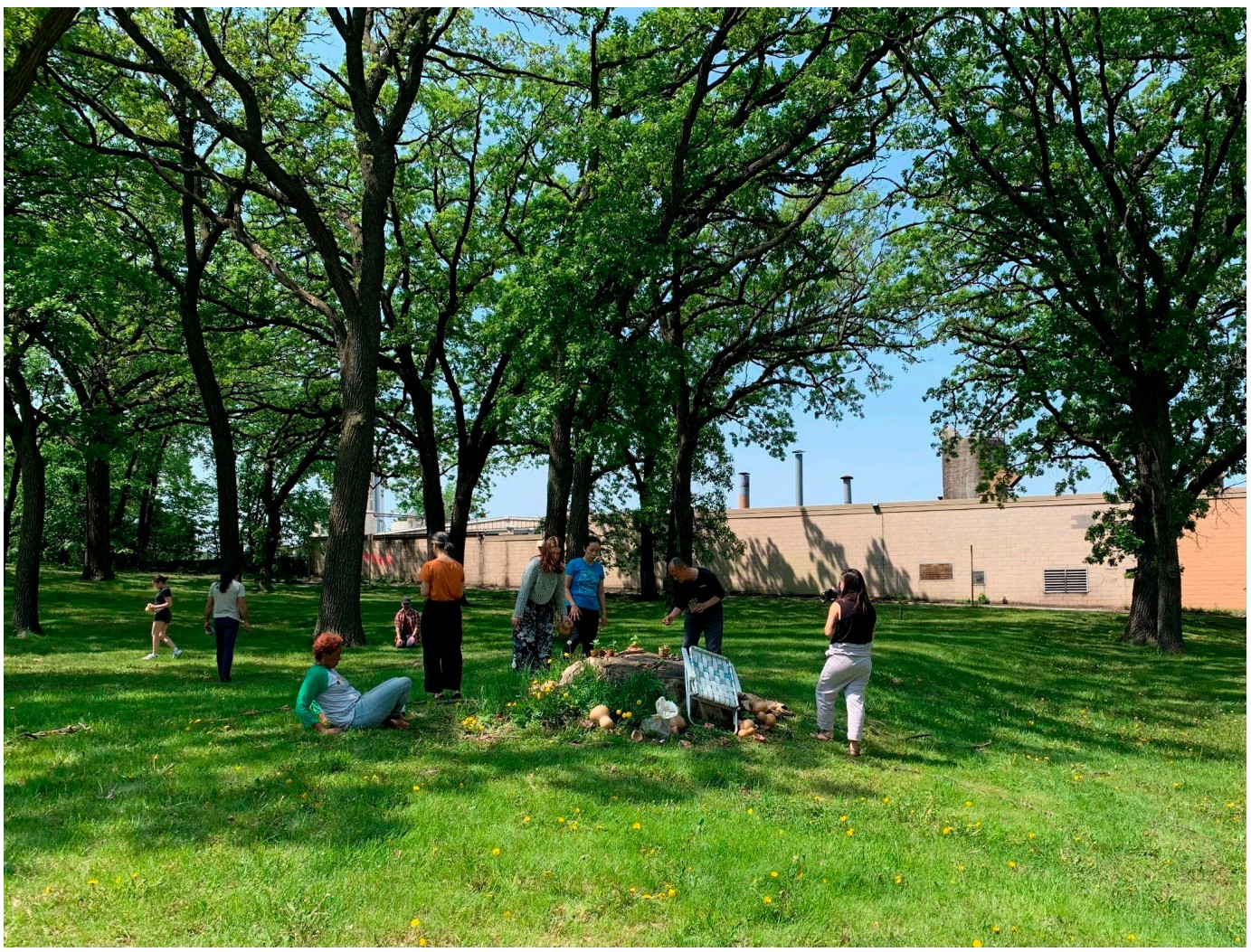

**Figure 5.** Prairie | Concrete work session, May 2023, Frogtown Farm, St. Paul, MN photo by Pramila Vasudevan.

## 3. Repatterning Modalities of Making through Prairie | Concrete

Prairie | Concrete (Vasudevan 2023) takes place at three park sites, including Western Sculpture Park[12], Frogtown Park & Farm[13], and Hidden Falls Regional Park[14]. All of these sites are situated within Imnizaska, where the white rock bluffs form what we now call St. Paul, Minnesota. The public-facing portion of this project is made up of three workshops and one performance gathering at each site location, totaling nine workshops and three event gatherings throughout the summer of 2023. The Aniccha Arts collaborative (Vasudevan 2004) has twenty BIPOC[15] collaborators, who both engage the public in these workshop activities as well as perform in the event gatherings. Most of this collaborative live or work in St Paul or have a relationship with the sites. Many of these artists are from the neighborhoods of these sites and the larger surrounding areas. The structure of the public-facing portion of this project is intended to engage participants from the site locations in a reciprocal conversation through workshop activities that center the plants of the place. This engagement structure is meant to contribute to the evolution of each performance score and their particularities at each site.

Western Sculpture Park is a two-acre green space filled with sculptures and plants that is the backyard and living space to a few encampments, and to many apartment complexes that house Black, Latinx, and other immigrant communities. Frogtown Park & Farm is a thirteen-acre urban space where growing food and supporting the community's wellness

is a priority. The project activities are located in what used to be a sand and gravel mine, which, over many decades, has been transformed by a grove of big oak trees, grass, and dandelions. This farm is sacred and important to this neighborhood, especially to people from vulnerable contexts in the area. Hidden Falls Regional Park is an access point to the urban wilderness, where the movement of the Mississippi River and the biodiversity of the forest come into focus, yet it also keeps the surrounding communities at a distance. This park is connected to Crosby Farm Regional Park[16], where you can see the Bdote[17], a sacred site for Dakota peoples where two rivers meet. It is also close to Fort Snelling[18], a site of genocidal efforts of the Dakota peoples by the US government.

Like plants growing through concrete, Prairie | Concrete (Vasudevan 2023) is an opportunity to grow through the hardness of capitalism, casteism, racism, and settler colonialism, with the co-creation and tending of a garden, embodied movement practice, intimate conversation, cooking and sharing meals, and other ways of caring for each other in the place we are in with the people we are in relationships with. How can this project be a chance to root ourselves in our own personal histories, while leaning into the way that the land forms us? Is it possible to learn the language of the land, by creating movement and improvised dances from the shapes and life cycles of the trees and plants around us in this urban ecology? How can this embodied movement practice happen in a community and in solidarity with others, and not from an isolated perspective, nor from an appropriation of cultural forms? How can the compositional structures of the dance respond to the place? How is the movement score at Western Sculpture Park inspired by the root structures and branches of the trees, the surfaces of the large sculptures, and the languages spoken by the communities that reside there? Can the refusal of participation in any of the activities by the Aniccha Arts team or participants be truly an accepted part of the process of creating and even performing? How can we center plants by dancing for, with, and around them, while simultaneously inviting communities into an experience of observation and witnessing, moving away from spectatorship?

These outdoor gatherings are not romantic getaways from the settler colonial harm that so many of us are complicit in; on the contrary, I hope we can engage in a real embodied conversation that takes us to crucial depths of listening and processing.

**Funding:** This research received no external funding.

**Data Availability Statement:** Not available.

**Conflicts of Interest:** The author declares no conflict of interest.

## Notes

[1] Aniccha Arts (est. 2004) is a transdisciplinary arts collaborative in Minneapolis/St. Paul, MN, that produces site-specific performances which examine agency, voice, and group dynamics within community histories, institutions, and systems.

[2] Richa Nagar, a friend and an academic who often invites me into her classroom, talks about how we can engage in a path of continuous (un/re)learning and in that way (un)becoming as a social justice practice of being in the world.

[3] Prairie | Concrete is an outdoor embodied movement project that will bring visibility to plant cycles and growing practices through embodied listening and movement sessions with communities in St. Paul, Minnesota. The commission will unfold across three City of St. Paul Public Parks—Frogtown Farm, Hidden Falls Regional Park, and Western Sculpture Park.

[4] Pillsbury House Theatre, originally a settlement house, is a human services and arts institution where I worked as a teaching artist with vulnerable communities and also worked as the program director for Naked Stages, their early career performance development program.

[5] The caste system is a South Asian social hierarchical structure of discrimination originated in Hinduism which is determined by a person's birth.

[6] In Habit: Living Patterns is a 75 min outdoor performance that was performed in loops continuously for nine hours under the Central Avenue bridge in Minneapolis. This piece is based on 16 "dots" or segments, commenting on different aspects of the collectively-learned habits and behaviors that emerge in everyday life. Each section was an interrogation of the patterns that exist in our bodies and environment.

[7] Upstream Arts is an institution that uses the power of the creative arts to activate and amplify the voice and choice of individuals with disabilities.

8    Every Other is an Aniccha Arts performance that examines how the body is disciplined through personal, material, spatial, and sociopolitical sites of tension. Featuring a multi-racial cast that moves through an immersive sonic and light environment, this interdisciplinary performance is composed of three movements that explore themes such as the relationship between body and space, the problem of cultural appropriation, and the implications of losing one's first language.

9    Census is an outdoor performance that featured 65 performers and hundreds of audience members moving together across 170 yards for 8.5 h at the all-night Northern Spark Festival in Minneapolis. The piece asked: How does systematic information gathering affect the body, our everyday lives, and our broader communities? The cast featured members of historically underrepresented communities especially in terms of race, ethnicity, dis/ability, and gender-nonconformity. By moving together from dusk till dawn, the performers activated and re-articulated this space into a new kind of census, a living document of our own communities.

10    Parking Ramp Project was a performance installation inside a seven-level parking garage in Bloomington, MN. The project asked questions about transience, migration, and stability in a space that temporarily stores cars and is home to nothing. Performers pervaded the parking structure with their bodies, working against the ramp's visible slant to find their individual verticality. Questions we asked in creating the work: How do we find softness in a landscape of concrete? What anchors us on these alternating planes? How do we connect across such a complex landscape?

11    Prairie | Concrete is is an outdoor embodied movement project commissioned by Public Art Saint Paul to take place in three public parks in St. Paul through the summer of 2023. This project will bring visibility to plant cycles and growing practices through embodied listening and movement sessions with surrounding communities.

12    Western Sculpture Park is a 2 acre neighborhood park located by the state capitol. Public Art Saint Paul curates and maintains the sculpture collection and presents programs in the park.

13    Frogtown Park & Farm was formed in 2013 as a partnership between the Trust for Public Land, the City of St. Paul and the Wilder Foundation. It was created as a natural area, a recreation area and an urban demonstration farm.

14    Hidden Falls Regional Park is connected to Crosby Farm by a trail system and borders the Mississippi river with a large biodiversity. There is a small waterfall within this park.

15    BIPOC stands for Black, Indigenous, People of Color. I use this term since I have met indigenous people who do not identify with the category of 'people of color' and this feels like the best option to describe the multiple identities of the Aniccha Arts team working on the Prairie | Concrete project.

16    Crosby Farm Regional Park is connected to Hidden Falls regional park, home to large biodiversity by the Mississippi river and where you can see the Bdote.

17    Bdote is a Dakota word referring to the confluence of two rivers, and in this case the Minnesota and MIssissippi rivers.

18    Fort Snelling is a former military fortification, a primary center for government forces during the US-Dakota war of 1862.

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
