# Peer review of "My Practice of Re-Patterning My Art"

_arts, 2023_

Round 1

Reviewer 1 Report

Thank you for sharing and tracing your creative process. This is very rich and valuable to the field. I see how past projects and their affiliated research have brought you to Prairie | Concrete. I think there is potential to see more writing towards the end about your current project, and then a clearer conclusion that ties the past and present together. 

Author Response

Thank you for your feedback.  I have taken your feedback into consideration and expanded the description of Prairie|Concrete leading to a better conclusion.  

Reviewer 2 Report

The essay needs further substantiation and formal rearrangement. As it stands, even if the topic is of interest to the ARTS journal area, it should not be accepted.

Author Response

I have flushed out the footnotes and reference list to improve the substantiation.  But, I understand that this may be still be inadequate.  Thank you for the time and consideration of this essay.

Round 2

Reviewer 2 Report

After the additions and effort made by the authors to strengthen the essay, I am of the opinion that, as it stands, the article can be published. 

Author Response

Thank you for accepting this essay.